# Biomanufacturing Biotinylated Magnetic Nanomaterial via Construction and Fermentation of Genetically Engineered Magnetotactic Bacteria

**DOI:** 10.3390/bioengineering9080356

**Published:** 2022-07-30

**Authors:** Junjie Xu, Shijiao Ma, Haolan Zheng, Bo Pang, Shuli Li, Feng Li, Lin Feng, Jiesheng Tian

**Affiliations:** 1State Key Laboratory of Agrobiotechnology, College of Biological Sciences, China Agricultural University, Beijing 100193, China; junjiexu89@163.com (J.X.); msj_91@cau.edu.cn (S.M.); zhenghaolan95@163.com (H.Z.); cbspb@cau.edu.cn (B.P.); lishuler@163.com (S.L.); 2School of Mechanical Engineering & Automation, Beihang University, Beijing 100083, China; 3College of Life Science, Huaibei Normal University, Huaibei 235000, China; rx2500@163.com

**Keywords:** biomaterials, magnetotactic bacteria, engineered magnetosome, biosynthesis, fermentation

## Abstract

Biosynthesis provides a critical way to deal with global sustainability issues and has recently drawn increased attention. However, modifying biosynthesized magnetic nanoparticles by extraction is challenging, limiting its applications. Magnetotactic bacteria (MTB) synthesize single-domain magnetite nanocrystals in their organelles, magnetosomes (BMPs), which are excellent biomaterials that can be biologically modified by genetic engineering. Therefore, this study successfully constructed in vivo biotinylated BMPs in the MTB *Magnetospirillum gryphiswaldense* by fusing biotin carboxyl carrier protein (BCCP) with membrane protein MamF of BMPs. The engineered strain (MSR−∆F−BF) grew well and synthesized small-sized (20 ± 4.5 nm) BMPs and were cultured in a 42 L fermenter; the yield (dry weight) of cells and BMPs reached 8.14 g/L and 134.44 mg/L, respectively, approximately three-fold more than previously reported engineered strains and BMPs. The genetically engineered BMPs (BMP−∆F−BF) were successfully linked with streptavidin or streptavidin-labelled horseradish peroxidase and displayed better storage stability compared with chemically constructed biotinylated BMPs. This study systematically demonstrated the biosynthesis of engineered magnetic nanoparticles, including its construction, characterization, and production and detection based on MTB. Our findings provide insights into biomanufacturing multiple functional magnetic nanomaterials.

## 1. Introduction

The heightened awareness of actively mitigating climate change, reducing pollution, and improving food security has driven the need for biomanufacturing. Biosynthesis provides a key platform for bioproduction [1], including green synthesis of biodiesel [2,3,4], ethanol [5,6], organic acids [7,8], and biopolymers [9]. Compared with those, fossil-based to bio-based material production represents a more significant challenge in biosynthesis technology and high-volume output [1].

Among various materials, magnetic nanomaterials display a higher specific area and increased magnetic, optical, and electrical stability and are widely applied in imaging [10,11], cancer treatment [12], biosensors [13], gene delivery [14], and agriculture sectors [15,16]. Traditional chemical synthesis of Fe_3_O_4_ nanoparticles (NPs) is performed by precipitating Fe^2+^/Fe^3+^ [17,18]. Organic solvents (e.g., methacrylic acid, esters, oleyl alcohol, and oleic acid) were used as stabilizers or surfactants in the manufacturing process, possessing severe adverse environmental effects and eventually causing harm to human health. Compared with traditional synthesis, green synthesis of nanomaterials using different biological entities, such as plant debris, and microorganism cells, overcomes many destructive effects of physical and chemical techniques [19,20,21]. These include synthesizing nanomaterials at mild pH, pressure, and temperature, without requiring toxic or hazardous substances and without adding external reducing, capping, and stabilizing agents [22,23]. However, modifying the synthesized NPs is challenging.

MTBs are environmental microorganisms that can synthesize single-domain magnetite (Fe_3_O_4_) nanocrystals in their organelles, magnetosomes (BMPs) [24,25,26]. BMP synthesis is strictly controlled by a set of genes [27]. Thus, BMPs display high chemical purity, a narrow size range, and consistent crystal morphology [28]. BMPs can be extracted through cell lysis and magnetic adsorption [29,30]. BMPs are biocompatible, without cytotoxic, genotoxic, immunotoxic, and hemolytic effects [31,32,33,34]. Compared with artificial NPs, BMPs have higher transverse relaxivity (r2) and more effective signal decay, and they can be used as magnetic resonance imaging (MRI) contrast agents [35,36]. Moreover, the outer membrane of BMPs makes them easily modified with drugs, antibodies, and nucleic acids, widely used in drug delivery, antigen recovery and detection, and gene transfection [37,38,39].

BMPs can be modified chemically and biologically. Chemical modification is performed by using a chemical crosslinking agent, coupling functional molecules to the BMP surface. However, most crosslinking agents are toxic and cannot ensure the activity of the molecules modified on the BMPs. Biological methods are the most efficient mode for BMPs modification. Recombinant MTB strains can be constructed using the BMP surface display technique to produce genetically engineered BMPs [40,41,42,43], which makes functional molecules directional and arranges them in order on the BMP surface, avoiding crosslinking [44]. However, the yield of recombinant strain and BMPs is low, approximately 0.59 ± 0.03 g/L and 14.8 ± 0.5 mg/L, respectively [45].

Biotin binds specifically with streptavidin/avidin, forming a biotin–avidin system (BAS). BAS is one of the most available tools for biological research, used in protein purification, protein localization analysis, and immobilization [46,47,48,49]. In this study, we adopted the BMP surface display technique to construct biotinylated BMPs in *Magnetospirillum gryphiswaldense*, the model strain for analyzing BMPs synthesis and bioproduction [50]. Moreover, we attempted to increase the throughput of engineered BMPs.

In this study, we successfully constructed a recombinant strain (MSR−∆F−BF) for biotinylated BMP (BMP−∆F−BF) production, evaluated the cell yield, and engineered BMPs in a 42 L fermenter. BMP−∆F−BF extraction conditions were developed and optimized. BMP−∆F−BF coupled with streptavidin was invested and compared with commercially and chemically constructed biotin-labeled beads.

## 2. Materials and Methods

### 2.1. Bacterial Strains and Culture Conditions

The plasmids, mutants, and bacterial strains used in this study are listed in Table 1. *E. coli* S17-1 BF harboring plasmid pBBR−bccp−mamF was used as a donor strain for biparental conjugation; it was cultured in Luria–Bertani (LB) medium containing kanamycin (50 μg/mL). The *mamF* mutant strain (MSR−∆F) was used as the recipient strain and cultured at 30 °C for 24 h in sodium lactate/ammonium chloride/yeast extract (LAY) medium containing gentamycin (5 μg/mL) and nalidixic acid (5 μg/mL), as described previously [40]. MSR-1 transformants termed MSR−∆F−BF were constructed through biparental conjugation and cultured in a LAY medium containing gentamycin (5 μg/mL), kanamycin (5 μg/mL), and nalidixic acid (5 μg/mL). Furthermore, the expression of target proteins was induced by adding isopropyl β-D-1-thiogalactopyranoside (95.32 μg/mL) during the logarithmic period, and the mixture was cultured for another 12 h.

The cell growth and magnetic response of MSR−∆F−BF were measured, and wild-type MSR-1 and MSR−∆F were used as controls. The cell density (OD_565_) and magnetic response (Cmag) were measured every 2 h using a UV–visible spectrophotometer in the bacteria culture process (UNICO2100; UNICO Instrument Co., Shanghai, China). MSR−∆F−BF cell growth and Cmag curves were constructed to detect the status of strain growth and magnetosome synthesis.

### 2.2. Construction of Recombinant Strains MSR−∆F−BF

The recombinant plasmids were constructed as shown in Figure 1A. The *mamF* and *bccp* genes were amplified from the genomic DNA of MSR-1 using corresponding primers. The initiation codon of *mamF* and termination codon of *bccp* were removed during amplification. *mamF* and *bccp* were then fused into *bccp-mamF* using fusion PCR, with a hydrophilic polypeptide linker (Gly_4_Ser)_3_ between fusion genes. *bccp-mamF* was cloned into a broad host-vector pBBR1MCS-2 to construct the recombinant plasmids pBBR−bccp−mamF. The pBBR−bccp−mamF was transformed into MSR−∆F by biparental conjugation to construct a recombinant strain, MSR−∆F−BF. Selecting MSR−∆F as the host strain can increase the amount of biotin displayed on engineered BMPs. The authigenic biotin ligase link biotin to a lysine residue of biotin carboxyl carrier protein (BCCP) displayed on BMP provided a recombinant magnetosome, BMP−∆F−BF. The MSR−∆F native *mamF* gene in the genome was replaced by the gentamycin-resistance gene. To detect the expression of fusion genes *bccp*−*mamF*, the *gfp*−marked expression plasmids pBBR−bccp−mamF-gfp were constructed. Western blotting was performed to evaluate the green fluorescent protein (GFP) expression level.

### 2.3. Transmission Electron Microscope (TEM) Observation

A TEM was used to observe the morphology of cell and magnetosomes of MSR−∆F−BF, MSR−WT, and MSR−∆F. The strains were cultured in LAY media for 24 h at 30 °C and collected by centrifugation at 12,000 rpm for 2 min at 4 °C. Pellets were rinsed three times with ddH_2_O. Cultured strains (1 mL) were suspended in 50 μL ddH_2_O. Then, 10 μL of the suspension was dropped onto a copper mesh for 10 min, dried for 30 min using infrared light, and observed under a TEM (model JEM-1230, JEOL; Tokyo, Japan).

Structural details of the magnetosomes in MSR−∆F−BF were determined using a high-resolution transmission electron microscope (HRTEM) at ESPCI (Paris, France) on a JEOL 2010F microscope at 200 kV, equipped with a Gatan cooled CCD camera (1024 × 1024 pixels). For each particle, the structural parameters of the crystals were determined from the two-dimensional fast Fourier transform of the HRTEM image. The magnetic properties of BMP−∆F−BF were determined using a vibrating sample magnetometer (Model 3900; Princeton Measurements Corporation, Princeton, USA, sensitivity 5.0 × 10^–10^ Am^2^) from the Paleomagnetism and Geochronology Laboratory.

### 2.4. Submerged Culture of M. gryphiswaldense MSR−∆F−BF

The recombinant strain *M. gryphiswaldense* MSR−∆F−BF was cultured in a 42 L fermenter (BioFlo 110; New Brunswick Scientific, New Jersey, USA). Cell and magnetosome yields were calculated, as described previously [44]. The inoculum (5 mL) stored at −80 °C was activated in a 50 mL LAY medium containing gentamycin (5 μg/mL), kanamycin (5 μg/mL), and nalidixic acid (5 μg/mL) at 100 rpm at 30 °C for 24 h. After subculturing thrice, 10% (*v*/*v*) inoculum was transferred to the 42 L fermenter for submerged culture.

The 42 L fermenter was filled with a 30 L culture medium containing 20 g (70–80%) sodium lactate, 5.0 g NH_4_Cl, 6.0 g MgSO_4_.7H_2_O, 15.0 g K_2_HPO_4_.3H_2_O, 15.0 g yeast extract, and 17.5 mL Wolfe mineral solution. The feeding medium contained 500 g lactic acid, 90 mL NH_3_.H_2_O, 6.0 g MgSO_4_.7H_2_O, 15.0 g yeast extract, 15.0 g K_2_HPO_4_.3H_2_O, 10.0 g FeCl_3_.6H_2_O, and 17.5 mL Wolfe mineral solution was dissolved in 700 mL ddH_2_O. The culture and feeding medium were sterilized for 30 min at 121 °C. Gentamycin (5 μg/mL), kanamycin (5 μg/mL), and nalidixic acid (5 μg/mL) were added before inoculation. The pH was maintained at 6.8 by automated supplementation of the feeding medium supplemented with high concentration of lactate, and the temperature was maintained at 30 °C. The initial airflow was 0.5 L/min, and the stirring rate was 100 rpm. Once the dissolved oxygen (dO_2_) decreased to 15%, we increased the airflow to 1 L/min and maintained dO_2_ between 0% and 1% by regulating agitation (10 rpm every 2 h). When the dO_2_ decreased to 5%, 7.5 mL 1M isopropyl β-d-1-thiogalactopyranoside (IPTG) was added to induce the *bccp-mamF* gene expression. The cell density and magnetic response were measured every 4 h during the fermentation process.

### 2.5. Extraction and Characterization of BMP−∆F−BF

Fermenter−cultured MSR−∆F−BF was collected by centrifugation at 10,000 rpm at 4 °C for 15 min. The collected cells were resuspended in 20 mM phosphate−buffered saline (PBS) with a mass volume ratio of 1:10 (pH 7.4) and disrupted using an ultrasonic homogenizer (Ningbo Scientz Biotechnology Co., Ltd, Ningbo, China) for 20 min at 3 s/time with a time interval of 5 s at 400 W to lyse the cells. BMPs were collected from the disrupted cell suspension using a columnar neodymium−boron magnet. The supernatant was discarded, and the separated magnetosomes were resuspended in 10 mM PBS (pH 7.4), sonicated for 15 min at 3 s/time with a time interval of 5 s at 150 W, and washed twice. Then, we reduced the power to 100 W and washed the sample several times until the supernatant protein concentration was lower than 0.1 mg/mL. The purified BMP−∆F−BF was resuspended in 25% glycerol and stored at 4 °C.

Purified BMP−∆F−BF and wild-type BMP were observed under a TEM. For this purpose, a small number of BMPs was suspended in 1 mL of deionized water and thoroughly dispersed by ultrasonication for 10 min. Then, 10 μL of the suspension was dropped onto a copper mesh for 10 min, air-dried, and observed under a TEM. Hydrated radii and zeta potential of BMP−∆F−BF were detected at Nanjing Dongna Biological Technology Co., Ltd., Nanjing, China.

### 2.6. Preparation of Biotin−Labeled BMPs with NHS−Biotin

BMPs are membrane-enclosed magnetite nanocrystals. The primary amines of the BMP membrane were biotinylated by NHS−Biotin to achieve biotin-labeled BMPs (BMP−biotin). Then, 2 mg wild-type BMP was suspended in 1 mL 10 mM PBS (pH 7.4), and 13.5 μL 20 mM NHS−Biotin was added. The reaction mixture was ultrasonicated (70 W) for 1 min, incubated with shaking (200 rpm) for 30 min at 25 °C, and placed over a magnet to isolate BMP−biotin. BMP−biotin was suspended in a 25% glycerol and stored at 4 °C.

### 2.7. Detection of the Streptavidin−HRP Linkage Rate of BMP−∆F−BF and BMP−biotin

The linkage rate of BMP−∆F−BF and BMP−biotin with HRP−streptavidin was compared with commercial biotin-labeled magnetic beads (MB−biotin; superparamagnetic iron oxide; particle size 20–50 nm; Nanocs, Natick, MA, USA). BMP−∆F−BF, BMP−biotin, and MB−biotin were incubated with 1% bovine serum albumin (BSA) and stored at 4 °C. One−step ELISA was used to detect the linkage rate. A 96−well microtiter plate coated with 300 μL 2% BSA was incubated at 4 °C overnight and washed thrice with 250 μL 10 mM PBST (pH 7.4). Then, 2 mg BMP−∆F−BF, BMP−biotin, and MB−biotin each were added (each BMPs and MB three parallel) and captured using a magnetic rack. The sample was then washed thrice with PBST, and 100 μL 1:2000 diluted HRP-streptavidin (Beijing Solarbio Science & Technology, Beijing, China) was added to each well, with coupling for 1 h at 600 rpm at room temperature, captured by a magnetic rack and washed five times with PBST. The color was developed using 100 μL tetramethylbenzidine (TMB) for each well, reacted for 3 min at room temperature, and then 50 μL 2 M H_2_SO_4_ was added to stop the reaction. The supernatant was transferred to a new 96−well microtiter. Absorbance at 450 nm was measured on a microplate reader (blank control, only BMPs and MB incubated, no HRP-streptavidin). The OD_450_ value was calculated to compare the BMPs and MB linkage rates.

## 3. Results

### 3.1. Construction of the Engineered Strain MSR−∆F−BF

Previously, we adopted the “BMP surface display technique” to exhibit the functional molecule (protein A) on BMPs [40] used for the detection or diagnostic immunoassays, having a high Ab-conjugation and antigen-adsorption capability [44]. Here, we fused BCCP expression with BMPs abundant membrane protein (MamF) to display biotin on BMPs (Figure 1A). Firstly, a fusion expression vector was constructed. The *bccp* and *mamF* genes were fused into the *bccp-mamF* gene by fusion PCR then cloned into a broad host-vector pBBR1MSC-2 to construct fusion expression vector pBBR−bccp−mamF. Then, pBBR−bccp−mamF was transformed into MSR−∆F by biparental conjugation to construct an engineered strain, MSR−∆F−BF. The authigenic biotin ligase can link biotin to a lysine residue of BCCP protein, obtained biotin-labeled engineered BMPs (BMP−∆F−BF). MSR−∆F native *mamF* gene in the genome was replaced by the gentamicin-resistance gene. Choosing MSR−∆F as the host strain can increase the amount of biotin displayed on engineered BMPs.

To detect the expression of the fusion gene, a *gfp*-labeled three-gene fusion expression vector was constructed. The expression of fusion protein BCCP-MamF-GFP on engineered BMPs was confirmed by Western blotting (Figure 1B and Appendix A). Compared with wild-type BMPs (BMP−WT), engineered BMPs displayed a specific band at 55 kDα. TEM observation indicated that no BMPs were synthesized in MSR−∆F, whereas new small-size BMPs were synthesized in MSR−∆F−BF (Figure 1C). The fusion gene expression complemented the disruption of the *mamF* gene in MSR−∆F−BF, confirming that the fusion gene was successfully expressed in MSR−∆F−BF.

The MSR−∆F−BF cell growth and BMP synthesis states were detected in the flask and compared with MSR−∆F and wild-type MSR-1. Flask-cultured MSR−∆F−BF growth was slower than that of MSR−WT but faster than that of MSR−∆F. OD_565_ reached the highest (~0.98) at 22 h, approaching MSR−WT (~1.01) at 20 h (Figure 1D). As no BMPs were synthesized in MSR−∆F, no Cmag value was detected. The highest Cmag value of MSR−∆F−BF was ~0.3 at 16 h, 3.33-fold lower than that of MSR−WT (~1.0) at 12 h (Figure 1E), because the BMP size in MSR−∆F−BF was smaller than that in MSR−WT. These results indicate that the constructed MSR−∆F−BF grows well and can synthesize BMPs. MSR−∆F−BF can be used for producing of biotinylated BMPs.

### 3.2. Magnetic Properties Analysis of MSR−∆F−BF

MSR−∆F−BF displayed a lower Cmag value, and the synthesized BMP size was smaller than that of MSR−WT. TEM showed that MSR−∆F−BF synthesized BMPs displayed sufficient spaces between the particles. In this section, we firstly adopted high-resolution TEM and fast Fourier transform to analyze the BMP composition of MSR−∆F−BF. The BMP−∆F−BF crystal shape remained a hexagonal octahedron, and a few possessed an irregular shape (Figure 2A). Despite the small size of BMP−∆F−BF, it exhibited a typical ferroferric oxide lattice structure.

To elucidate the magnetic properties of BMPs in MSR−∆F−BF, the hysteresis loops were measured (at room temperature) and compared with that of MSR−WT. Hysteresis loops for MSR−WT displayed a bigger gap between two lines than that for MSR−∆F−BF (Figure 2B). The coercivity and saturation magnetization values for MSR−∆F−BF were 2.24 and 4.26, respectively, much smaller than those for MSR−WT (7.65 and 12.55, respectively) (Table 2). Compared with MSR−WT, MSR−∆F−BF synthesized small-size BMPs, which decreased its magnetic properties, but the core constituent was still noted as Fe_3_O_4_. This property displays better applicability, and the dispersibility of BMP−∆F−BF will be better than that of BMP−WT. Subsequent experiments will attest to this viewpoint.

### 3.3. Submerged Culture of MSR−∆F−BF in Bioreactor

Large-scale cultivation of the engineered strains is the basis for engineered BMP application. However, MTB large-scale cultivation is challenging, especially the cultivation of engineered strains. Similarly, MTB such as *Magnetospirillum gryphiswaldense* MSR-1, *Magnetospirillum magnetotacticum* MS-1, *Magnetospirillum magneticum* AMB-1, *Magnetovibrio blakemorei* MV-1, and *Desulfovibrio magneticus* RS-1 can only be cultivated on the laboratory scale. The BMP production level is low. The order of the BMP yield is MSR-1 > AMB-1 > MS-1 [53]. Previously, Zhang et al. achieved the highest cell yield of 9.16 g/L and BMP yield of approximately 356.52 mg/L by optimizing the MSR-1 fed-batch culture feeding medium [54]. Later, we reported the yield cultivation of an engineered strain (*M. gryphiswaldense* ∆F-FA) on a large scale and achieved the highest cell and BMP dry weight of 2.26 g/L and 62.29 mg/L, respectively [44].

We attempted to culture *M. gryphiswaldense* MSR−∆F−BF in a 42 L fermenter and detected the cell and magnetosome yield. BMP synthesis began when dO_2_ decreased to ~1% (Figure 3A). Low dO_2_ concentration inhibited cell growth; however, high dO_2_ concentration influenced BMP synthesis. Thus, dO_2_ concentration control is significant for MSR-1 fermentation. The gradual increase in dO_2_ caused a significant increase in biomass. Therefore, we controlled the dO_2_ value between 0% and 1% by regulating the airflow and agitation speed to ensure bacterial growth and BMP synthesis. Cmag reached 1.3 after 48 h of culturing and then began to decline. MSR−∆F−BF fermentation was complete when the Cmag value decreased to ~0.8 (Figure 3B). The maximum cell and BMP yield were 8.14 g/L and 134.44 mg/L, respectively, ~3-fold higher than those of *M. gryphiswaldense* ∆F-FA. These findings indicate that MSR−∆F−BF is potentially useful for industrial fermentation processes, although further optimization of culture conditions is necessary.

### 3.4. Preparation and Characterization of BMP−∆F−BF

Fermented cells were stored at −80 °C and used for engineered magnetosome purification. Cells at 1 g wet weight were suspended in 10 mL PBS, high-power ultrasonication was applied to disrupt the cell, and BMP−∆F−BF was separated using a magnet (Figure 4A). Then, BMP−∆F−BF was washed several cycles by a low-power ultrasonic bath to remove cell debris on BMPs. After washing six times, the supernatant protein concentration was lower than 0.1 mg/mL and nearly stable, indicating that no associated proteins dropped from BMP−∆F−BF (Figure 4B). TEM showed the extracted BMP−∆F−BF; no protein fragment was observed in the background, providing purified engineered BMPs (Figure 4C).

The dispersibility of nanomaterial in solution played a key role in applications. As mentioned above, the engineered strain MSR−∆F−BF synthesized small-sized BMPs and displayed weak magnetism compared with wild type. Therefore, BMP−∆F−BF showed good dispersibility in the solution. First, we measured the purified BMP−∆F−BF particle size using the ImageJ software (National Institutes of Health, Bethesda, MD, USA). The average particle size was 20 ± 4.5 nm, smaller than that of MSR−WT (35 ± 7.8 nm). BMP−∆F−BF and BMP−WT displayed a narrow diameter distribution (Figure 4D). Then, the hydrated radii and zeta potential of BMP−∆F−BF in a water-based suspension system were detected and compared with BMP−WT and commercial MB−biotin (Table 3). The BMP−∆F−BF hydrated radii value was 53.52 ± 10.23 nm, much smaller than the wild type (113.9 ± 62.19 nm) and MB−biotin (669.3 ± 288.5 nm). The zeta potential reflected the particle surface charge, with a value above +30 mV or below −30 mV indicating good particle dispersity. BMP−∆F−BF’s zeta potential was −23.6 ± 6.12 mV, close to −30 mV, lower than wild type (−17.7 ± 6.78 mV) and MB−biotin (−17.5 ± 3.84 mV). These results indicated that the smaller particle sizes and lower BMP–∆F–BF magnetic properties exhibit outstanding dispersity in water–based suspension systems and render them suitable candidates for further BMP–∆F–BF applications.

### 3.5. Detection of BMP−∆F−BF Coupling Streptavidin Rate

The linkage rate of BMP−∆F−BF to streptavidin was detected and compared with BMP−biotin, constructed through chemical modification and commercial MB−biotin. First, the conditions for the linkage between BMP−∆F−BF and streptavidin were optimized, including the coupling time, buffer, and BMP−∆F−BF and streptavidin weight rate (Figure 5A). The bonding between biotin and avidin displayed the strongest affinity (Kd = 10~15 M), leading to the linkage time (Figure 5A1) and buffer (Figure 5A2) having no significant influence on the streptavidin linkage rate. BMP−∆F−BF (0.1 mg) was used to bind with different amounts of streptavidin (ranging from 0.1–0.5 mg) to optimize the linkage weight rate of BMP−∆F−BF and streptavidin (Figure 5A3). As streptavidin’s weight increased, BMP−∆F−BF linkage with streptavidin increased, and the streptavidin nonspecific absorption of BMP−WT increased. When the BMP−∆F−BF and streptavidin weight rate was 1:3, the rate of BMP−∆F−BF linkage with streptavidin was considerably higher than BMP−WT nonspecific absorption and 1:1 and 1:2 conditions, indicating suitable conditions for BMP−∆F−BF linkage with streptavidin.

BMP−∆F−BF linkage with the streptavidin rate was compared with chemically modified BMP−biotin. The schematic diagram of BMP−biotin construction is shown in Figure 5B. BMP in vitro biotinylation was constructed by incubating with NHS-biotin. The -NHS group of NHS-biotin can react with the -NH_2_ group of the BMP membrane and modify biotin onto BMP, forming biotin-labeled BMP. BMP−∆F−BF linkage with the HRP-labeled streptavidin (Streptavidin−HRP) rate was compared with those of BMP−biotin and commercial MB−biotin (Figure 5C). BMP−∆F−BF and BMP−biotin linkage with streptavidin rates were almost the same, lower than MB−biotin. However, as the storage time increased to 75 d, BMP−biotin and MB−biotin linkage with the Streptavidin−HRP rate markedly decreased, and the BMP−∆F−BF linkage rate became more stable. One possible explanation is that nonspecific chemical modification helped the reaction with all amino groups on BMP. Biological modification is specific to the BMPs transmembrane protein (e.g., MamF) as an anchoring protein, ensuring stable protein expression on BMPs.

## 4. Discussion

Interest in nanomaterials and especially nanoparticles has exploded in the past decades primarily due to their novel or enhanced physical and chemical properties compared to the bulk material. These extraordinary properties have created a multitude of innovative applications in the fields of medicine and pharma, electronics, agriculture, chemical catalysis, the food industry, and many others. More recently, nanoparticles are also being synthesized “biologically” through the use of plant− or microorganism−mediated processes, as an ecofriendly alternative to the expensive, energy−intensive, and potentially toxic physical and chemical synthesis methods [55]. Among these organisms, MTB are unique as they synthesize magnetic nanoparticles in their organelles (magnetosome) under strict and fine genetic control [24,25,26].

Magnetosomes are excellent biomaterials, but only a few MTBs could achieve tractability and straightforward cultivation because of their restricted living conditions. Only two strains in the genus *Magnetospirillum*—*M. gryphiswaldense* MSR−1 and *M. magneticum* AMB-1—could be cultivated in bulk. Unfortunately, the magnetosome productions of these strains were much lower than the chemical synthesized magnetic nanoparticles [29], and the productions by those engineered stains were even lower [44]. In this study, we used MSR-1 as a host strain for the bioproduction of biotinylated magnetosomes; the engineered magnetosomes yield ~134.44 mg/L, over twofold higher than the latest report by another recombinant strain [44]. In addition, there are several advantages to this recombinant strain and bioengineered magnetosomes. First, the strain MSR−∆F−BF was cultivated in a 42 L fermenter, while stains in most other reports (including wild−type and recombinant strains) were cultivated in a shaking flask or 5 L fermenter [29], which implied that MSR−∆F−BF could be cultivated more suitably at an industrial scale. Second, in terms of the size of the bioengineered particle, BMP−∆F−BF (20 ± 4.5 nm) was smaller than BMP−WT (35 ± 7.8 nm), which meant better dispersity in the water−based suspension system. Third, compared with commercial biotin−labeled magnetic beads and chemically constructed biotinylated magnetosomes, BMP−∆F−BF has better storage stability.

The process of magnetosome biotinylation in this work could also be considered an effective biotin surface display technique. The surface display is a recombinant technology that expresses target proteins on the surface of biomembranes, biostructures, or even biomolecules, and can be applied to almost all types of biological entities from viruses to mammalian cells. Phage display, mRNA display, ribosome display, bacteria display, and yeast display are among the most commonly used methods. These techniques have been used for various biotechnical and biomedical applications such as therapy of disease, drug screening, biocatalysts, library screening, quantitative assays, and biosensors [56,57,58]. However, most functional molecules expressed on the surface of display platforms are proteins or peptides. The successful display of biotin on the magnetosome surface suggested the possibility and strategies display biomolecules beyond proteins by bioengineering methods, which would greatly increase the applicable scopes of the above systems.

Up until now, biotin has been the only nonprotein biomolecule employed in the magnetosome surface display system. Biotin is widely distributed in animal and plant tissues and has a molecular weight of 244.31 Da with two ring structures: imidazolone ring, the main site noncovalently binding with avidin/streptavidin or special receptors with high specificity and affinity [59]; and thiophene ring, with the pentanoic acid as the side chain at C2, whose carboxyl terminal is the only structure to bond covalently with other biomolecules [60]. Biotin carboxyl carrier proteins (BCCPs) are molecules of 69~73 amino acid residues, to which a biotin group is covalently attached through a lysine residue [61]. The avidin/biotin couple is popular in the conjugation of nanosystems. Biotinylated nanoparticles were widely used for biological research and biomedical applications, including the sensing of analytes or the delivery of drugs [62,63]. In this work, fusion expression of mamF and bccp genes resulted in magnetosome surface modification by functional biotins in vivo, without grafting and conjugation steps of artificial nanoparticles.

Particularly, bioengineered biotinylation of magnetosomes brings an important new function to the display system. It is known that biotin receptors are overexpressed in different tumor cells, including breast, ovarian, and lung cancer [64,65]. Thus, the approach of biotin functionalization has been explored as an attractive strategy for tumor nuclear magnetic resonance imaging, targeted-drug, and gene−delivery systems [66,67,68,69]. Consequently, BMP−∆F−BF could be an excellent potential candidate as an imaging agent and targeted-delivery carrier for tumors.

In brief, we systematically introduced the bioengineering and biofabrication process of biotinylated magnetosomes. Both the strategy and resulting engineered magnetosome could be promising examples for preparing multiple functional biogenic magnetic nanomaterials.

## Figures and Tables

**Figure 1 bioengineering-09-00356-f001:**
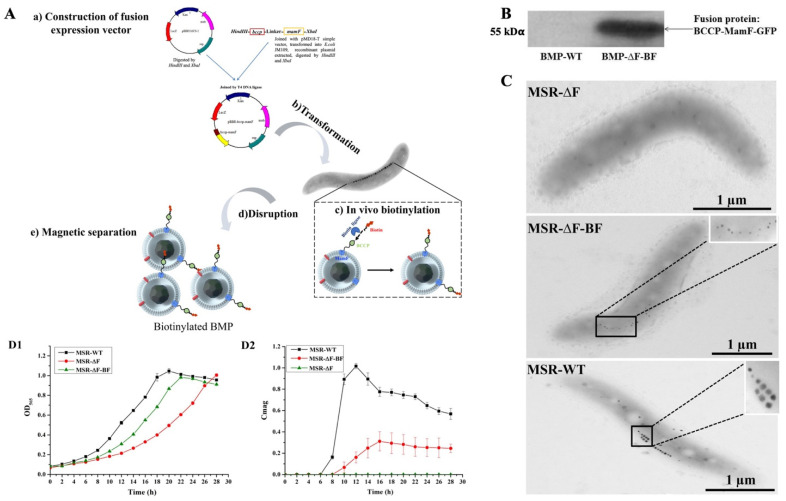
Construction and growth of the engineered bacteria MSR−∆F−BF and BMP synthesis detection. (**A**) Schematic diagram of MSR−∆F−BF construction and biotinylated BMP extraction. (**a**) construction of *bccp* and *mamF* genes fusion expression vector (pBBR−bccp−mamF); (**b**) transformation of pBBR−bccp−mamF to MSR−∆F by biparental conjugation to construct the engineered strain (MSR−∆F−BF); (**c**) in vivo biotinylated BMP by biotin ligase linkage of biotin to BCCP on BMP; (**d**) ultrasonic homogenizer-disrupted cell; (**e**) magnetically separated biotinylated BMP from cell debris. (**B**) Western blotting detects the fusion protein display on engineered BMP. (**C**) TEM micrograph of MSR−∆F, MSR−∆F−BF, and MSR−WT; there were no BMPs in MSR−∆F cell, small−sized BMPs in MSR−∆F−BF cell, and normal particle size BMPs in MSR−WT cell. The results indicated that the fusion gene was successfully expressed and complemented the disruption of the *mamF* gene in MSR−∆F−BF, suggesting MSR−∆F−BF could synthesize of BMPs successfully. (**D**) Cell growth curve (**D1**) and Cmag curve (**D2**); MSR−∆F−BF cell growth rate was close to MSR−WT and faster than MSR−∆F. Cmag value was lower in MSR−WT.

**Figure 2 bioengineering-09-00356-f002:**
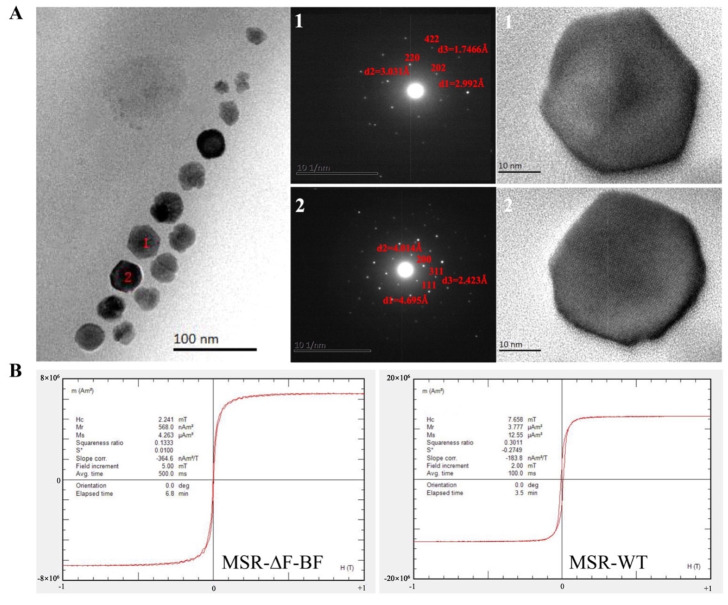
(**A**) High−resolution TEM and fast Fourier transform analysis of the crystal composition of BMP−∆F−BF; the core constituent of BMP−∆F−BF was Fe_3_O_4_ NPs. (**B**) Magnetic property analysis of MSR−∆F−BF, S*: static moment.

**Figure 3 bioengineering-09-00356-f003:**
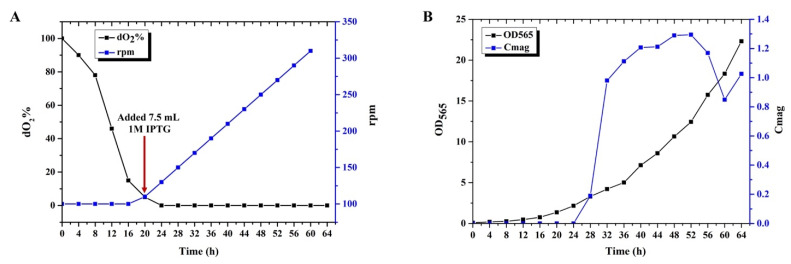
Submerged culture of MSR−∆F−BF in a 42 L fermenter (**A**) rpm and dO_2_% values as a function of time. As the cell grew, the dO_2_% in the broth was consumed continuously. dO_2_% decreased to ~5%, and IPTG was added to introduce fusion gene expression. dO_2_% decreased to ~1%; the agitation speed was added at 10 rpm/2 h to maintain microaerobic condition. (**B**) MSR−∆F−BF cell growth and Cmag value curve. The highest OD_565_ and Cmag reached ~22 and ~1.3, respectively.

**Figure 4 bioengineering-09-00356-f004:**
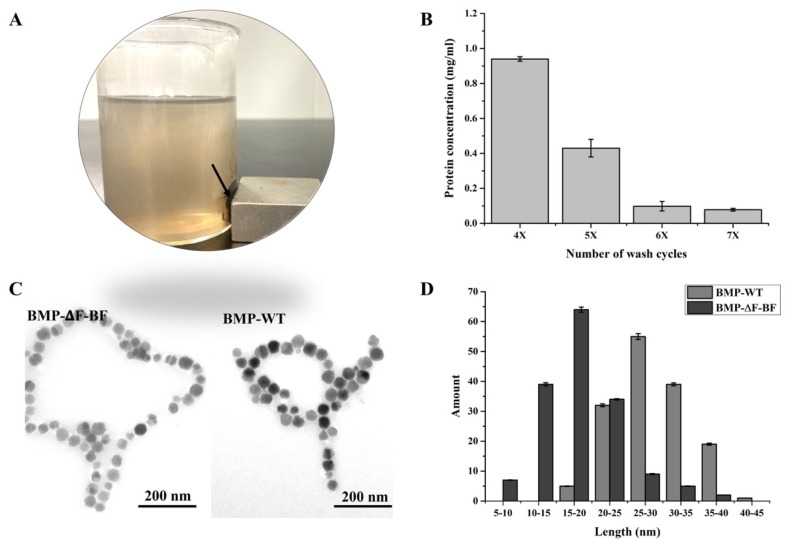
Purification of BMP–∆F–BF. (**A**) Magnetic–rack–separated BMP–∆F–BF (arrow). (**B**) Protein concentration in the supernatant following various numbers of wash cycles. After washing six times, the protein concentration was lower than 0.1 mg/mL, and as the wash cycle increased, the protein concentration stopped decreasing. (**C**) TEM micrograph of purified BMP–∆F–BF after washing six times. There was no stain on the photo background, and BMP–∆F–BF was well purified. (**D**) Particle size distribution of BMP–∆F–BF and BMP–WT.

**Figure 5 bioengineering-09-00356-f005:**
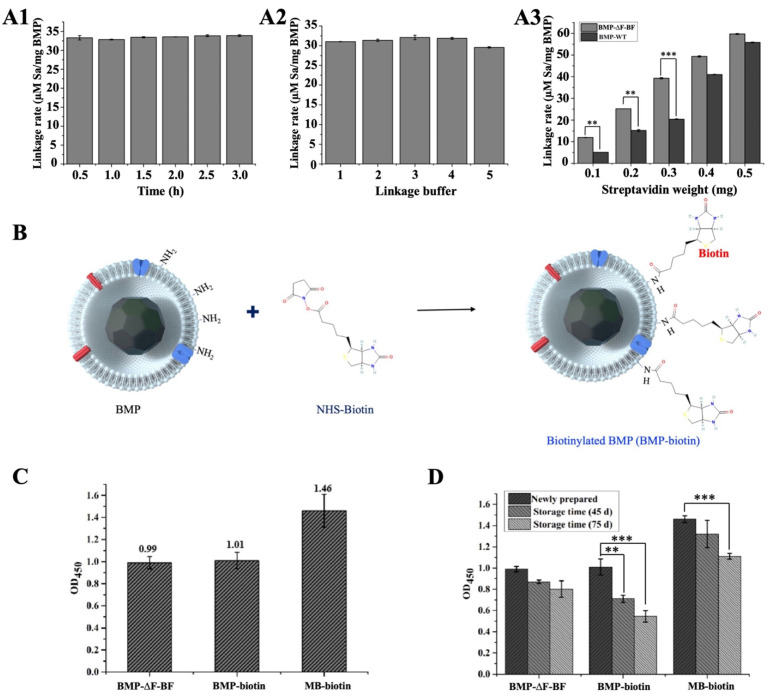
Comparison of streptavidin linkage rate with BMP−∆F−BF, BMP−biotin, and MB−biotin. (**A**) Optimization of conditions for conjugation of streptavidin to BMP−∆F−BF: (**A1**) linkage temperature, (**A2**) linkage buffer, 1:10 mmol/L Tris-HCl, pH 7.4, 2:10 mmol/L Hepes, pH 7.4, 3:10 mmol/L Gly, pH 7.4, 4:10 mmol/L Na_2_HPO_3_-C_6_H_8_O_7_, pH 7.4, 5:10 mmol/L PBS, pH 7.4; (**A3**) streptavidin weight. Linkage temperature and buffer have no significant influence on the streptavidin linkage rate of BMP−∆F−BF. When BMP−∆F−BF and streptavidin weight rate was 1:3, BMP−∆F−BF linkage with streptavidin rate was markedly higher than BMP−WT nonspecific absorption (**A3**), which were suitable conditions for BMP−∆F−BF linkage with streptavidin. (**B**) Schematic of chemically conjugated biotin to BMP and BMP co-incubated with NHS-biotin. The -NHS group can react with -NH_2_ group of BMP modified streptavidin to the surface of BMP, forming biotin-labeled BMP. (**C**) Comparison of Streptavidin−HRP linkage rate of BMP−∆F−BF, BMP−biotin, and MB−biotin. (**D**) Comparison of Streptavidin−HRP linkage rate of BMP−∆F−BF, BMP−biotin, and MB−biotin at different storage times. The biosynthesized BMP−∆F−BF linkage rate was more stable than that of BMP−biotin and MB−biotin. Data are presented as the mean  ±  s.d. (*n*  =  3 biological replicates per group) and statistically analyzed using the two-sided Student’s *t*-test: ** *p* < 0.01, *** *p* < 0.001.

**Table 1 bioengineering-09-00356-t001:** Strains and plasmids used in this study.

Strain and Plasmid	Description	Source or Reference
Strains		
*Magnetospirillum gryphiswaldense* MSR-1	Wild-type magnetotactic bacteria, Nx^r^	DSM 6361
*Magnetospirillum gryphiswaldense* MSR−∆F	*mamF*-defective mutant, Nx^r^ Gm^r^	Ref [40]
*Magnetospirillum gryphiswaldense* MSR−∆F−BF	*mamF* mutant harboring pBBR−bccp−mamF, Nx^r^ Gm^r^ Km^r^	This study
*E. coli* DH5α	*endA1 hsdR17*(r_K_^−^ m_K_^+^) *supE44 thi-1 recA1 gyrA* (Nal^r^) *recA1* ∆(*lacZYA-argF*)*U169 deoR*[Φ80d*lacZ*∆M15]	Ref [51]
*E. coli* S17-1	t*hi endA recA hsdR* with RP4-2-Tc::Mu-Km::Tn*7* integrated into the chromosome; Sm^r^	Ref [52]
Plasmids		
pMD18-T simple	Clone vector, Amp^r^	Takara
pBBR1MCS-2	Broad-host-range shuttle vector, Km^r^	Lab collection
pBBR−bccp−mamF-gfp	pBBR1MCS-2 containing *bccp-mamF-gfp*, Km^r^	This study
pBBR−bccp−mamF	pBBR1MCS-2 containing *bccp-mamF*, Km^r^	This study

**Table 2 bioengineering-09-00356-t002:** Coercivity and saturation magnetization values of MSR−∆F−BF and MSR−WT.

Strains	Coercivity (Hc, mT)	Saturation Magnetization (Ms, µ Am^2^)
MSR−∆F−BF	2.24	4.26
MSR−WT	7.65	12.55

**Table 3 bioengineering-09-00356-t003:** Hydrated radii, zeta potential and polydispersity of BMP–∆F–BF, BMP–WT and MB–biotin.

Magnetosomes/ Magnetic Beads	Hydrated Radii (nm)	Zeta Potentials (mV)	Polydispersity
BMP–WT	113.9 ± 62.19	−17.7 ± 6.78	0.48
BMP–∆F–BF	53.52 ± 10.23	−23.6 ± 6.12	0.35
MB–biotin	669.3 ± 288.5	−17.5 ± 3.84	0.59

## Data Availability

The data that support the findings of this study are available from the first author (J.X.) upon reasonable request.

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
