# Peer review of "Biomanufacturing Biotinylated Magnetic Nanomaterial via Construction and Fermentation of Genetically Engineered Magnetotactic Bacteria"

_bioengineering, 2022, doi:10.3390/bioengineering9080356_

Round 1

Reviewer 1 Report

The goal of the study was the use of the constructed recombinant strain( MSR-∆F-BF) for biotinylated BMPs (BMP-∆F-BF) production, evaluation of the cell yield, and the up-scaling the BMPs in a 42-L fermenter. The authors developed and optimized BMP-∆F-BF extraction conditions. They  examined and compared  BMP- ∆F-BF coupling streptavidin rate   with commercial and chemical constructed biotin-labeled beads. It is a very interesting piece of work which may be interested to the readers. The article does not bring scientifing novelty but the subject is important from scientific point of view. The experiments were well designed, the content of the state of art provides useful information about the topic. As was mentioned by the authors the results of the study showed that the maximum cell and BMPs yield was higher than M. gryphiswaldense ∆F-FA. The results of the work indicated that MSR-∆F-BF may potentially be useful for industrial fermentation processes. The authors underlined that further optimization of culture conditions is necessary. Sumarizing, the manuscript is well-written, it is logically structured, the results are interpreted well and supported by appropriate references or arguments. However Engish should be revised throuhgout. The authors should send the paper to the native speaker. My recommendation is to accept the article for the possible publication in “Bioengineering“ after the revision.

Author Response

Reviewer 1

Point 1: The goal of the study was the use of the constructed recombinant strain( MSR-∆F-BF) for biotinylated BMPs (BMP-∆F-BF) production, evaluation of the cell yield, and the up-scaling the BMPs in a 42-L fermenter. The authors developed and optimized BMP-∆F-BF extraction conditions. They  examined and compared  BMP- ∆F-BF coupling streptavidin rate   with commercial and chemical constructed biotin-labeled beads. It is a very interesting piece of work which may be interested to the readers. The article does not bring scientifing novelty but the subject is important from scientific point of view. The experiments were well designed, the content of the state of art provides useful information about the topic. As was mentioned by the authors the results of the study showed that the maximum cell and BMPs yield was higher than M. gryphiswaldense ∆F-FA. The results of the work indicated that MSR-∆F-BF may potentially be useful for industrial fermentation processes. The authors underlined that further optimization of culture conditions is necessary. Sumarizing, the manuscript is well-written, it is logically structured, the results are interpreted well and supported by appropriate references or arguments. However Engish should be revised throuhgout. The authors should send the paper to the native speaker. My recommendation is to accept the article for the possible publication in “Bioengineering“ after the revision.

Response: Thank you for your nice comments; English editing was finished with the help of Editage. The editing certificate has been sent to the editor.

Reviewer 2 Report

Biomanufacturing of biotinylated magnetic nanomaterial via construction and fermentation of genetically engineered magnetotactic bacteria

The successful construction of in vivo biotinylated BMPs is interesting and compiling. Please see specific comments below. 

  1. The introduction can be expanded: specifically, more details should be given on traditional synthesis vs bio-synthesis. Successful methods should be compared in throughput, efficiency, and contamination to explain why biosynthesis is a good direction to develop in the future. 
  2. Table 1: three lines still say Ref - please add references to this. 
  3. Figure 1: B. Please include full gel in SI. The figure caption has to be more detailed for part c to explain the difference between each image and what you want the readers to see.
  4. Figure 2: red label in A 1.2. are unclear, and also needs to show reproducibility. 
  5. Figure 4: needs to better explain the experimental set-up, what is the difference expected between df and WT, why does the difference matter, and what is the result? Part D needs to be reproduced with error bars. 
  6. Figure 5: A1 and A2 need statistic analysis and p-value. coupling buffer 1-5 needs to be better explained.

Reviewer 3 Report

This manuscript entitled "Biomanufacturing of biotinylated magnetic nanomaterial via 2 construction and fermentation of genetically engineered mag-3 netotactic bacteria" is a good scientific work focused on the bioproduction of magnetic stable nanoparticles. The paper is reasonably well-written but some changes are advisable, as follows:

1) Intro: please, indicate the most relevant features of BMPs that make then an ideal material. What means ideal in this context?

2) Materials&Methods: please, indicate the supplier, purity and other key characteristics of the reference material: commercial biotin-labeled magnetic beads (MB-biotin).

3) Magnetic properties analysis of MSR-ΔF-BF. The authors describe these properties; however, even if a short discussion is added at the end of the manuscript, some explanation of the results is needed. Moreover, there is no comparison to the reference material (commercial biotin-labeled magnetic beads (MB-biotin)).

4) Detection of BMP-ΔF-BF coupling streptavidin rate. It is good that the authors describe this rate by means of an equation; usually rate has to do with time of operation or processing. However, only absorbance at a given wavelength is provided in Figure 5C. Moreover, how can the authors explain the effect of particle size? Moreover, I wonder why the authors have chosen magnetic particles that are notably bigger than the BMPs for comparison.

5) Discussion. Is too short to discuss the results presented in the previous sections. It should be enlarged as this section is not really a Conclusion section.

Minor changes:

1) Avoid separating figures and tables from their captions between adjacent pages to facilitate reading and for aesthetic reasons.

2) Please, complete, even adding the statement "non applicable" when this is appropriate,  all the subsections at the end of the manuscript.

3) Try to provide clearer figures. For example, graphs A1, A2 and A3 in Figure 5 are difficult to see.

3) Minor English changes:

Line 46: "be controlled" instead of "controlled"

Line 50: "widely" instead of "wildly"

Line 52: please, write again the first sentence of the paragraph. It does not make sense.

Line 58: "strain" instead of "stain"

Round 2

Reviewer 3 Report

The authors have reasonably addressed all my comments and made the suggested corrections. This manuscript version is notably enhanced.

Author Response

Thank you for your nice comments. The English language has been revised.